# AneuG-Flow: A Large-Scale Synthetic Dataset of Diverse Intracranial Aneurysm Geometries and Hemodynamics

**Wenhao Ding**
Department of Biomedical Engineering
Imperial College London
United Kingdom
w.ding23@imperial.ac.uk

**Yiying Sheng**[*]
Department of Biomedical Engineering
National University of Singapore
Singapore
yiyings@nus.edu.sg

**Simão Castro**
Department of Mechanical Engineering
Instituto Superior Técnico
Portugal
simao.vitoriano@tecnico.ulisboa.pt

**Hwa Liang Leo**
Department of Biomedical Engineering
National University of Singapore
Singapore
bielhl@nus.edu.sg

**Choon Hwai Yap**
Department of Biomedical Engineering
Imperial College London
United Kingdom
c.yap@imperial.ac.uk

## Abstract

Hemodynamics has a substantial influence on normal cardiovascular growth and disease formation, but requires time-consuming simulations to obtain. Deep Learning algorithms to rapidly predict hemodynamics parameters can be very useful, but their development is hindered by the lack of large dataset on anatomic geometries and associated fluid dynamics. This paper presents a new large-scale dataset of intracranial aneurysm (IA) geometries and hemodynamics to support the development of neural operators to solve geometry-dependent flow governing partial differential equations. The dataset includes 14,000 steady-flow cases and 730 pulsatile-flow cases simulated with computational fluid dynamics. All cases are computed using a laminar flow setup with more than 3 million cells. Boundary conditions are defined as a parabolic velocity profile with a realistic waveform over time at the inlet, and geometry-dependent mass flow split ratios at the two downstream outlets. The geometries are generated by a deep generative model trained on a cohort of 109 real IAs located at the middle cerebral artery bifurcation, capturing a wide range of geometric variations in both aneurysm sacs and parent vessels. Simulation results shows substantial influence of geometry on fluid forces and flow patterns. In addition to surface mesh files, the dataset provides volume data of velocity, pressure, and wall shear stresses (WSS). For transient cases, spatial and temporal gradients of velocity and pressure are also included. The dataset is tested with PointNet and graph U-Nets for WSS prediction, which showed relative L2 loss of 4.67% for normalized WSS pattern.

---

[*]The first and second authors contributed equally to this work.

39th Conference on Neural Information Processing Systems (NeurIPS 2025) Track on Datasets and Benchmarks.

# 1 Introduction

For many vascular diseases, there has been a long history of investigating fluid mechanics behaviors aimed at improving surgical decision-making and post-operative treatment (1). However, a significant gap remains between biomechanics research and its adoption in clinical practice. A clear example of this gap is intracranial aneurysm (IA)—a condition where a weakness in the vessel wall leads to a bulge that carries a potentially lethal risk of rupture (2). Despite extensive research in biomechanics and several meta studies suggesting a correlation between biomechanical markers and aneurysm instability (3), these markers have yet to be accepted or utilized by physicians. In practice, physicians continue to rely primarily on morphological markers (4) to assess rupture risk, even though such systems (5) have been reported to suffer from high false-positive rates (up to 56%) (6).

One of the key reasons for this gap is that fluid dynamics simulations are time-consuming and require specialized expertise. As a result, physicians are unable to obtain fluid dynamics data in sufficiently large sample sizes to evaluate the clinical utility of biomechanics markers, preventing their adoption of it. As such, it would be useful to develop deep learning models for rapid prediction of flow dynamics in disease morphologies. Driven by this clinical demand, the concept of predicting CFD solutions directly from vascular geometries has gained significant momentum in recent years. In terms of application domains, models have been developed for coronary arteries (7), aorta (8), aortic aneurysms (9), and the left ventricle (10). From a methodological perspective, both supervised and self-supervised models are actively being explored. Representative work in the supervised category includes a family of geometry-informed neural operators (10; 11; 12; 13), while self-supervised approaches—such as (14; 15; 16)—leverage physics-informed training. Both approaches require a substantial dataset for training and/or validation.

Unfortunately, available datasets in the bioengineering domain to support such tasks are far from satisfactory. Most existing studies—particularly those focused on theoretical model development—rely on idealized geometry datasets. For instance, simple harmonic functions are employed in (15) and (16) to represent the 3D geometries of coronary arteries. Similarly, models in the geometry-informed neural operator family are often evaluated on highly simplified and impractical geometric domains (11), limiting their translational relevance to real-world applications. This limitation largely stems from the inherent difficulty of assembling a sufficient range of patient-specific geometries. As shown in Table 1, most available datasets contain only a few hundred geometries. Moreover, generating the corresponding CFD solutions demands significant High-Performance Computing (HPC) resources and considerable human expertise to design, tune, and maintain an efficient simulation workflow.

In addition to being limited in scale, several existing datasets suffer from other notable shortcomings. For example, (17) provides real aneurysm sac geometries, but attaches them to a single idealized mother vessel, resulting in globally unphysiological vascular structures with no variation. Similarly, while (18) offers a large-scale dataset, over 95% of them are manually deformed from real shapes rather than being generated through data-driven approaches, and the parent vessels were not generated even though they are critical to flow behaviors. To address the shortage of realistic 3D hemodynamics data and to offer a more physiological and large-scale alternative, we introduce a new dataset of IA geometries with associated hemodynamics. The main contributions are summarized as follows:

- We provide the first large hemodynamics dataset of both IA 3D geometry and its detailed hemodynamics where both aneurysm pouch and parent vessels are modelled. Geometries show decent diversity representative of that of clinical data.

- A relatively large amount of geometries are provided compared to existing datasets in the bioengineering community. We include 14,000 steady cases and 730 pulsatile cases.

- In addition to the volume data, we provide wall shear stress (WSS) solution on IA sac surfaces with a consistent mesh graph structure (connectivity), allowing easier downstream deep learning processing.

# 2 Case Description

## 2.1 Geometry

Unlike existing datasets which create geometry variations by manually warping real shapes (9; 18), we use a generative model AneuG (19) that learns the geometry distribution from a real IA cohort.

Table 1: Reivew on hemodynamics datasets

| Dataset | Anatomical region | Size | Geometry source |
|---|---|---|---|
| Faisal et al.(9) | Aortic aneurysm | 230 | Manually generated from 23 real shapes. |
| AnXplore(17) | Intracranial aneurysm | 101 | Real IA sac geometries merged with the same idealized mother vessel. |
| Aneumo(18) | Intracranial aneurysm | 10,000 | 9,534 synthetic shapes manually generated from 466 real shapes. |
| AneuG-Flow (Ours) | Intracranial aneurysm (with parent vessels) | 14,000 | Synthetic shapes generated with a generative model trained on a cohort of 116 real shapes. |

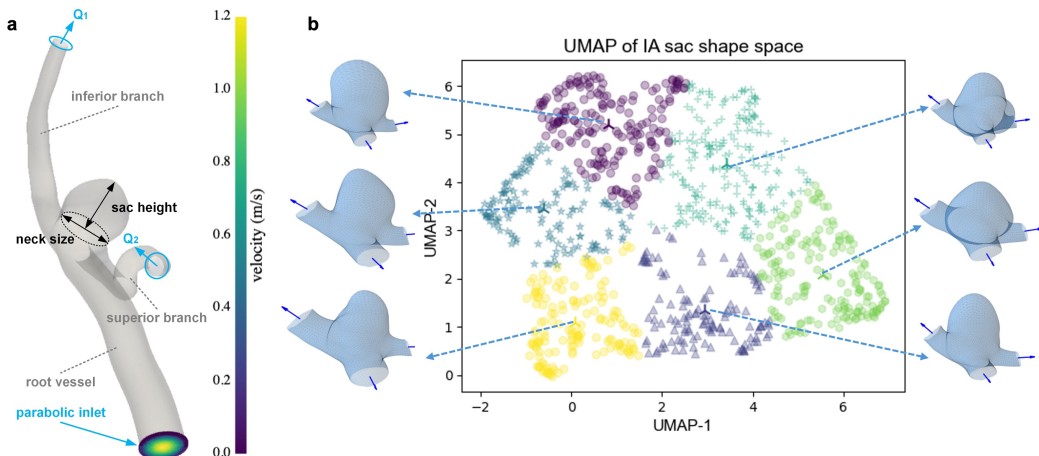

Figure 1: Geometry configuration. a: Flow configuration. b: UMAP of aneurysm sac geometries.

AneuG leverages the technique of Graph Harmonic Deform (GHD) (20) to encode the spatial warping of real IA shapes with respect to a canonical shape into a sequence of tokens. The distribution of tokens are then approximated with a two-stage Variational Autoencoder (VAE). More details can be found in (19) and (20). AneuG generates both the aneurysm sac and its parent vessels, with the latter being conditioned on the former. This approach allows us to create diverse physiological IA shapes for CFD simulations, while existing works such as (17) fails to capture the joint distribution of aneurysm sacs and their parent vessels. AneuG uses data from the AneuX morphology database (21), an open-access, multi-centric database combining data from three European projects: AneuX project, @neurIST project and Aneurisk. As reported in (21), all patients/participants provided written informed consent to participate in the study.

As shown in Fig. 1, we create IAs located at the bifurcation of the middle cerebral artery (MCA). Such a topology contains one root vessel and two downstream vessel branches (referred to as the inferior and superior branch). Compared to the inferior branch, the superior branch generally has smaller radius and larger spatial angle with respect to the root vessel.

## 2.2 Mesh Generation Pipeline

We assembled a set of open-source repositories and commercial software automation scripts to develop a fully automated pipeline for volume mesh generation. Low-resolution surface meshes generated from AneuG were processed using Geomagic Wrap v2024.3.0 for remeshing and localized smoothing. We then used the Vascular Modeling Toolkit (VMTK) to generate 3D volume meshes (codes modified from (8)). For each case, 4 inflation layers were applied with a total thickness equal to 0.5 times the local edge length near the wall. The final meshes contained an average of 3.4 million volume cells, which mesh convergence studies have shown to be sufficient (see supplementary

materials for details). This pipeline is generalizable to vascular structures with a fixed number of inlets and outlets. These automation codes can be found in (22).

## 2.3 Boundary Conditions

As shown in Fig. 1, we apply a parabolic velocity profile at the inlet with an average velocity of $0.684\,\mathrm{m/s}$, as measured in (24). For the outlets, several studies have demonstrated that flow split conditions more accurately represent physiological hemodynamics compared to fixed pressure boundaries (25; 26). Following these work, we determine the mass flow split between the superior and inferior branches using a modified form of Murray's law:

$$\frac{Q_1}{Q_2} = \left(\frac{D_1}{D_2}\right)^{\gamma}$$
(1)

where $Q_1$ and $Q_2$ are the outlet flow rates, $D_1$ and $D_2$ are the corresponding vessel diameters, and $\gamma$ is the flow split exponent. We follow (27) and choose $\gamma = 2.45$. This formulation ensures that the outlet boundary conditions reflect geometry-dependent mass flow split.

Blood were assumed as imcompressible non-Newtonian fluid. We adopt the Carreau–Yasuda model to account for the non-Newtonian behavior of blood:

$$\mu(\dot{\gamma}) = \mu_{\infty} + (\mu_0 - \mu_{\infty}) \left[1 + (\lambda\dot{\gamma})^a\right]^{\frac{n-1}{a}}$$
(2)

The model parameters are set as follows: zero-shear viscosity $\mu_0 = 0.056\,\mathrm{Pa\cdot s}$, infinite-shear viscosity $\mu_{\infty} = 0.00345\,\mathrm{Pa\cdot s}$, time constant $\lambda = 3.313\,\mathrm{s}$, and power-law index $n = 0.3568$. The Yasuda parameter is assumed to be $a = 2$ and the blood density is set to $1050\,\mathrm{kg/m^3}$.

An laminar flow setup was used, as the average Reynolds numbers in the root vessel and downstream branches were approximately $410$ and $330$, respectively. And vessel walls were assumed as rigid with a no-slip condition. Since measurements of blood pressure and flow velocity are not routinely performed during the clinical management of intracranial aneurysms, an average waveform reported in (28) was adopted for all transient simulations. The average inlet velocity was kept same across all morphologies to ensure that aneurysms with larger parent vessels experienced higher mass flow rates. The velocity waveform signal is included in our dataset. Each transient case was simulated over two cardiac cycles using a time step of $0.001, \mathrm{s}$, which was confirmed to be sufficient through a time-step convergence study. Only the results from the second cycle were extracted.

## 2.4 Solver and HPC Setup

All simulations were performed using ANSYS Fluent 2023R2 (ANSYS Inc., Canonsburg, PA, USA) on the High-Performance Computing (HPC) Services at Imperial College London and National University of Singapore. Simulations ran on AMD nodes, each contains 128 cores and 1TB RAM. The Research Data Store at Imperial College London were used for data storage during the runs. Each simulation was run on a single AMD node using 64 cores. Each case took approximately 3 minutes to mesh, and 2 minutes to solve the steady simulation. Transient cases each took around 10 hours to solve.

## 2.5 Graph structural consistency during CFD data extraction

In addition to the raw CFD solution data, we also provide additional post-processed WSS on the surfaces of IA sacs as graphs. Each graph contains the same number of nodes and the same connectivity, allowing easier implementation of downstream deep learning tasks. AneuG gnerates surface meshes using a mesh morphing approach, where every case has exactly 3500 triangle faces for the aneursym sac (19). As this mesh size is of low resolution, we subdivided each triangular element by adding a new node at the center of the each edge and dividing the element into four new elements, leading to 14,000 faces. WSS at nodes were then extracted through k-NN interpolation method from [23]. Further subdivision can be performed if higher resolution is desired. The standardization of node and connectivity structure allows a natural and easy node-to-node / edge-to-edge registration between different IA cases, and facilitates deep learning processing. The associated graph connectivity is also provided (see Table 2). Extraction codes are available at (22).

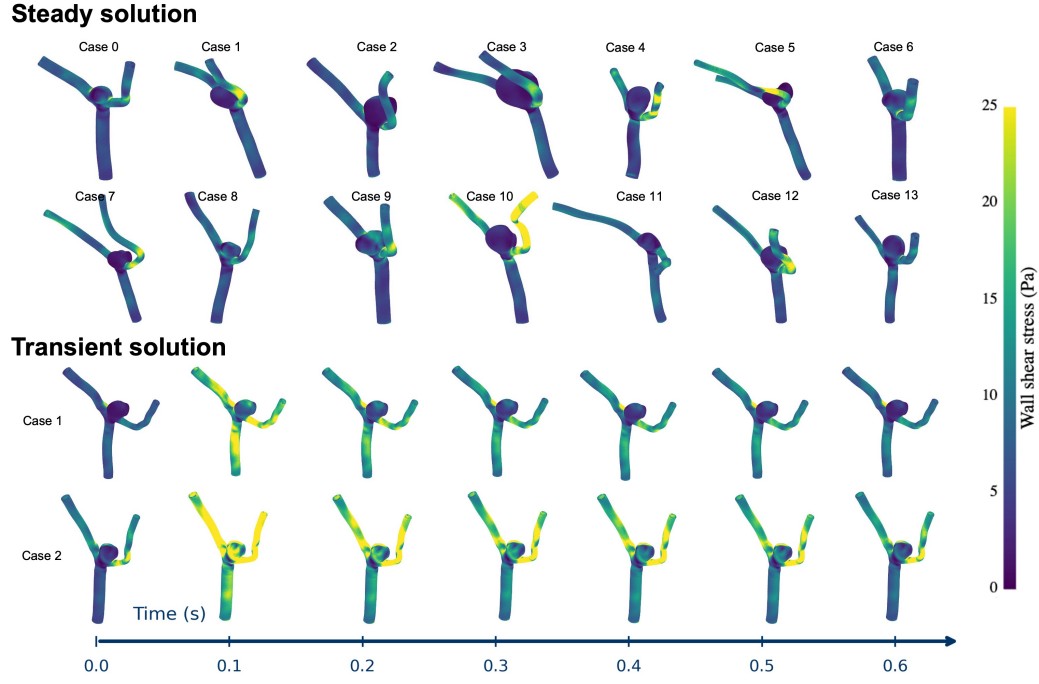

Figure 2: Example of wall shear stress data.

## 3 Dataset Description

### 3.1 Geometry Variation

We follow (19) to generate synthetic shapes using pretrained checkpoint files. Latent features were sampled from a 64-dimensional uniform distribution with a mean of 0 and a standard deviation of 2.5, and were subsequently passed through the decoder. We chose this setup because it covers over 98% of the latent space of a theoretically well-trained VAE. It is also the maximum deviation from the latent space center that still yields physiologically plausible shapes. As shown in Fig. 1 b, diverse aneurysm sac geometries are included.

### 3.2 Dataset Contents

A summary of the content of this dataset is provided in Table 2 and Table 3. For steady cases, we assemble the solution data of all cases into one Pytorch .pth file while keeping geometry-dependent files in separate case folders. As mentioned above, we also provide node-to-node registration for the aneurysm sac region to construct a well-structured PyTorch tensor object. This tensor has the shape of $[B, N, C]$, where $B$, $N$, and $C$ denote the number of cases, the number of surface nodes, and the number of physics variables, respectively.

We also provide a list of downsampled node indexes and associated edge connection of downsampled meshes obtained using the method described in (30). This downsampling approach preserves the topological integrity of the shapes, whereas generating low-resolution k-NN graphs based solely on Euclidean coordinates may introduce connections between points that are close in space but distant in terms of surface geodesic distance. These graph structures can be used for U-Net-like structures with graph convolution layers. A visualization is provided in Fig. 3.

For transient cases, we provide time-series data over one full cardiac cycle, as summarized in Table 3. Solutions were extracted at 80 uniformly spaced time steps within the second cardiac cycle, resulting in each PyTorch tensor object having the shape $[T, N, 1]$, where $T$ denotes the number of time steps and $N$ the number of nodes. PyTorch .pth files were saved separately within each case folder.

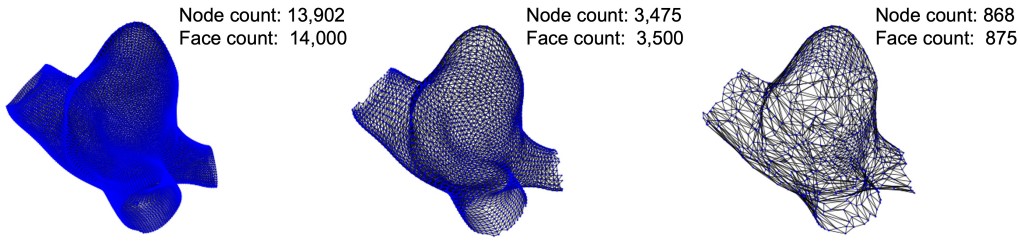

Figure 3: Mesh downsampling.

Table 2: Dataset contents of steady cases

| Solution data | | |
|---|---|---|
| **File & Structure** | **Keys** | **Content** |
| Raw_data.pth List[Dict[str, Any]] | label | List of solution variable names. |
| | tensor | PyTorch tensor object of solution data. |
| | ghd | GHD tokens (19) and rigid alignment checkpoints. |
| | log | Fluent residuals log. |
| | case_name | Case name. |
| Assembled_data.pth Dict[str, Any] | tensor | PyTorch tensor object of registered solution data. |
| | tensor_norm | Mean and standard deviation of registered data. |
| | idx_list | Vertex index list of downsampled surface meshes. |
| | edge_index_list | Edge connections of downsampled surface meshes. |
| | ds_factors | List of downsampling factors. |
| Geometry data (inside each case folder) | | |
| flowsplit_ratio.txt | | Mass flow split ratio between two outlets. |
| shape.obj | | Surface mesh generated by AneuG. |
| shape_remeshed.obj | | Post-processed surface mesh (by Geomagic). |

### 3.3 How to access the dataset

The **AneuG-Flow** dataset is open-source under a CC BY-SA 4.0 license. It can be accessed and downloaded directly from the Hugging Face Hub. Codes for processing the raw CFD data and training the baselines are available at GitHub (23).

### 3.4 Limitations

We aim to provide a large-scale hemodynamics dataset to support the prediction of biomechanical markers directly from patient-specific vascular geometries. However, several limitations remain. First, the dataset includes a relatively small number of transient cases (730) compared to that of steady cases (14,000), which may limit its utility for predicting temporal biomechanical markers such as time-averaged wall shear stress and oscillatory shear index. Expanding the dataset with more transient simulations is a priority for our future work. Second, the current dataset is limited to geometries with a single inlet and two outlets. To improve generalizability, additional vascular topologies — such as shapes with a single inlet and a single outlet — should be incorporated in the next phase. Finally, we do not consider variations of boundary conditions at the moment, as measuring blood pressure and mass flow waveform signals in the middle cerebral artery is not yet a routine in clinical practice.

Table 3: Dataset contents of transient cases

| File & Structure | Keys | Content |
|---|---|---|
| *Solution data (inside each case folder)* | | |
| blood_data.pt
Dict[str, Tensor] | various | Volume solution data of the blood domain, including spatial coordinates, velocity components, pressure, viscosity, temporal derivative of pressure, and spatial derivatives of velocity components. |
| wall_data.pt
Dict[str, Tensor] | various | solution data of the surface domain, including spatial coordinates, Wall shear stress components, and the total wall shear stress magnitude. |
| *Geometry data (inside each case folder)* | | |
| same as Table 2 | | |

## 4 ML Evaluation

We demonstrate a simple regression task as an application of the dataset. Given the geometries of IAs, we train several baseline models to predict the steady-state WSS map on the aneurysm sac. Leveraging the surface node-to-node registration described in Section 2.5, we construct input graphs with the same connectivity across different cases, each containing 13,902 nodes and 14,000 triangles. The network is designed to output WSS vector components in the $x$, $y$, and $z$ directions. We use 80% of the steady-state cases for training and the remaining 20% for testing.

**Networks & loss function design.** We adopt a U-Net-like structures for the models. Specifically, we train a PointNet++ and three graph U-Nets: one using simple Graph Convolutional Networks (GCN), another using Graph Attention Networks (GAT), and a third using Chebyshev Spectral Graph Networks (ChebNet) as the core convolutional layers. For PointNet++, downsampling is performed using farthest point sampling (FPS) (29). For graph U-Nets, a pre-computed set of downsampled node indices and associated edge connections was used, as mentioned in Section 3.2. A visualization of the mesh downsampling is provided in Fig. 3. In addition to the mean squared error (MSE) loss computed on z-score-normalized wall shear stress (WSS), we also evaluate an MSE loss defined on WSS values normalized using an power mapping:

$$\mathcal{L}_{\exp} = \mathrm{MSE}\left[f(w^{\mathrm{pred}}),\ f(w^{\mathrm{true}})\right], \quad f(w) = \left(\frac{\alpha \cdot w}{w_{\max}}\right)^{\beta} \cdot \frac{1}{\alpha} \tag{3}$$

where $w$ denotes the WSS components and $f$ is the normalization function. $\alpha$ and $\beta$ are manually selected hyperparameters. For this task, we chose $\alpha = 100$ and $\beta = \frac{2}{3}$. We introduce this loss term because high WSS values often appear near the junction areas between the aneurysm sac and its parent vessels. However, it is the low WSS distribution on the sac that is generally considered more clinically significant by physicians (3). By applying such a nonlinear normalization to the WSS, the model is encouraged to learn the overall spatial pattern of WSS rather than being dominated by high-magnitude areas. As shown in Fig. 4, the normalized WSS exhibits reduced contrast between high and low values, thereby emphasizing the underlying distribution pattern.

**Metrics & Results.** Each model is trained on a single NVIDIA RTX 3090 GPU for 24 hours, with the learning rate decayed by a factor of 0.75 every 100 epochs. Model performance is evaluated using the root mean square error (RMSE) and mean absolute error (MAE) computed on the WSS values. In addition, both metrics and the relative L2 error are computed on WSS normalized using Eq. (3), which reflects the model's capability of predicting the spatial pattern of WSS. Prediction performances are visualized for three random cases in Fig. 4. The model captures the global WSS map on the aneurysm sac well. As expected, high WSS is observed near the junction between the sac and the parent vessels, while low WSS appears around lobulated regions on the sac. Among different network designs, the U-Net using ChebNet as the convolutional layer performs the best. During training, the ChebNet kernel size was set to 3, and all networks were configured with identical depth

Table 4: Model performance evaluated on WSS and Eq. (3)-normalized WSS.

| Model | WSS | | Normalized WSS | | |
|---|---|---|---|---|---|
| | RMSE (Pa) | MAE (Pa) | RMSE | MAE | Relative L2 (%) |
| PointNet++ | 0.204 | 0.122 | 0.0353 | 0.0263 | 5.39 |
| U-Net (GCN) | 0.199 | 0.114 | 0.0317 | 0.0233 | 4.82 |
| U-Net (GAT) | 0.213 | 0.120 | 0.0333 | 0.0242 | 5.04 |
| U-Net (ChebNet) | 0.191 | 0.108 | 0.0307 | 0.0223 | 4.67 |

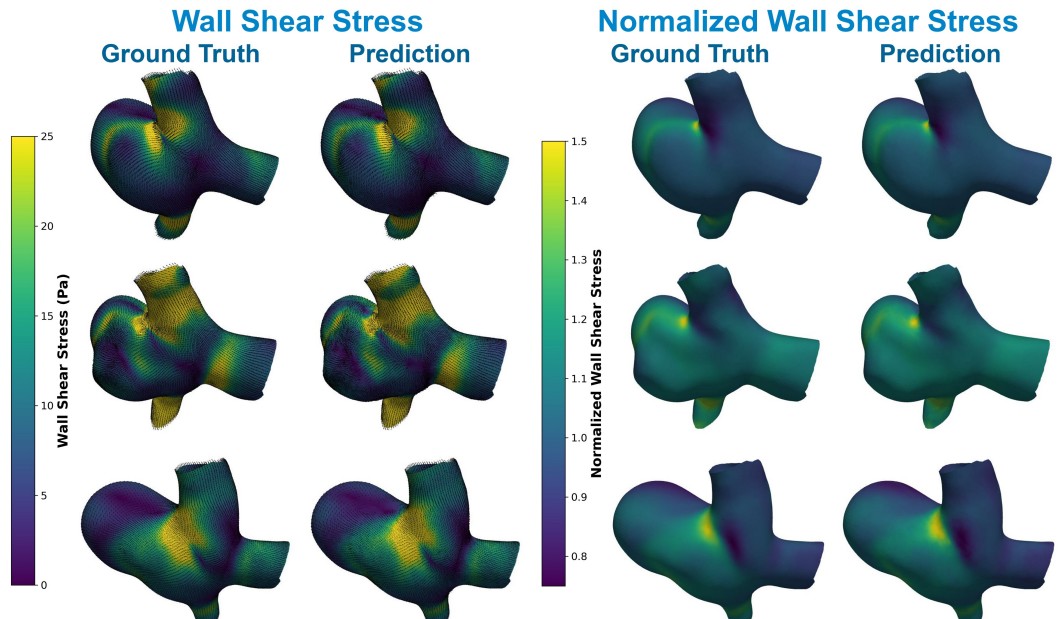

Figure 4: Prediction of WSS and WSS normalized with Eq. (3) from the trained U-Net (GCN).

and channel dimensions. The better performance is likely attributable to the enhanced receptive field of ChebNet, allowing more effective propagation of information across the mesh.

## 5   Relationship between morphological & biomechanical markers

We go beyond steady cases and investigate the relationship between morphological and biomechanical markers with our transient data. For morphological markers, we include Aspect Ratio (AR), Size Ratio (SR), and maximum sac height ($H_{max}$) as they are generally accepted in existing physiology research (4; 5). Further, We include Lobulation Index (LI) defined as the sac's surface area divided by its volume. We also compute the radius of a sphere with the same volume of the sac as it reflects the 3D size of the aneurysm, which we refer to as Equivalent Radius (ER). A detailed graphic definition of these markers can be found in the supplementary material. For biomechanical markers, we consider two of them: Oscillatory Shear Index (OSI) and Relative Residence Time (RRT) (31). OSI measures the directional changes of wall shear stress throughout a cardiac cycle, defined as:

$$\text{OSI} = \frac{1}{2} \times \left( 1 - \frac{|\int_0^T \boldsymbol{\tau_w}(t) dt|}{\int_0^T |\boldsymbol{\tau_w}(t)| dt} \right) \tag{4}$$

with values ranging from 0 (unidirectional flow) to 0.5 (oscillatory flow). Here $\boldsymbol{\tau_w}(t)$ is the wall shear stress vector. Relative Residence Time (RRT) combines both parameters to evaluate flow stagnation:

$$\text{RRT} = \frac{1}{(1 - 2 \times \text{OSI}) \times \text{TAWSS}} \tag{5}$$

where Time-Averaged Wall Shear Stress (TAWSS) is the average magnitude of wall shear stress over a complete cardiac cycle, calculated as:

$$\text{TAWSS} = \frac{1}{T}\int_0^T |\boldsymbol{\tau_w}(t)|dt \qquad (6)$$

It is generally considered dangerous when large OSI and RRT are observed on the aneurysm sac's surface. As shown in Fig. 5 and Fig. 6, most morphological markers demonstrated weak or statistically insignificant correlations with both OSI and RRT averaged on the aneurysm sac. This suggests that traditional morphological markers alone may be insufficient to capture the complex hemodynamic behavior within aneurysmal sacs.

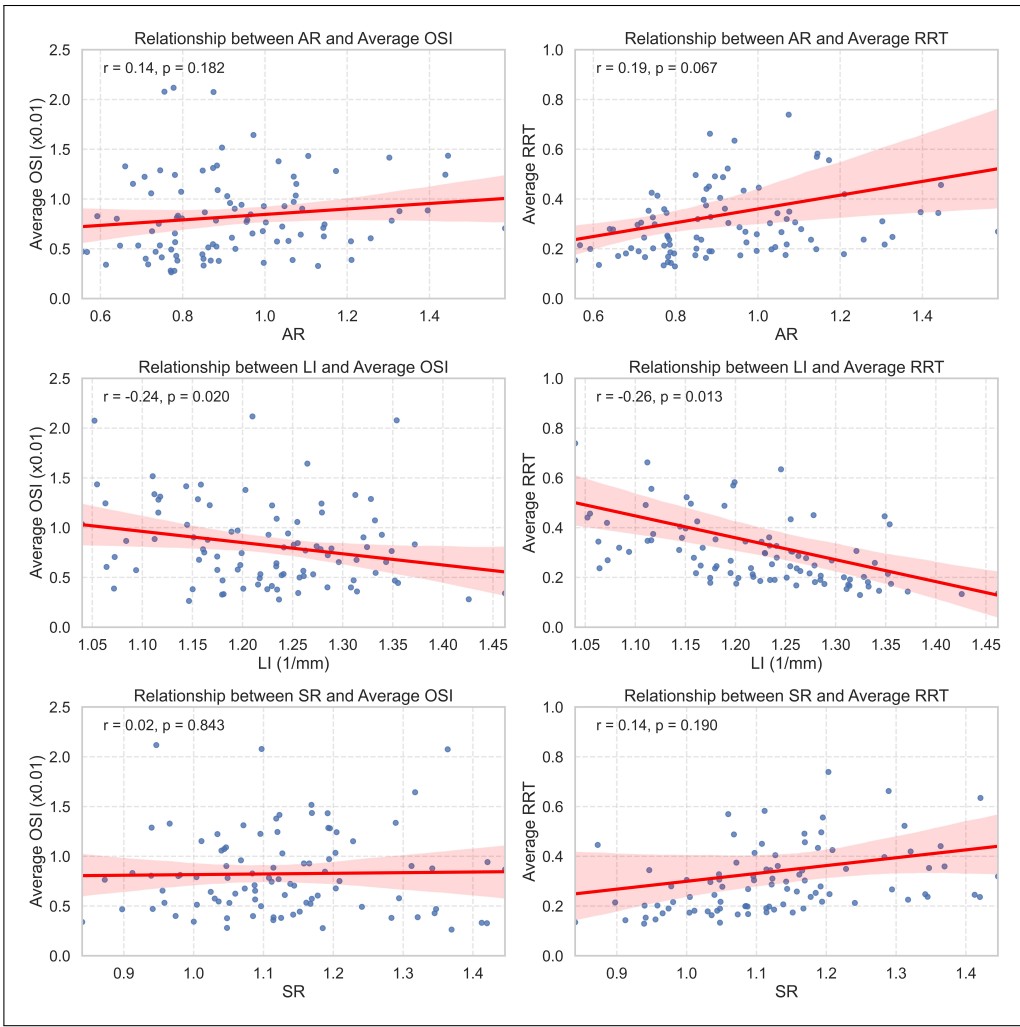

Figure 5: Morphological–biomechanical marker correlations (AR, SR, LI).

An interesting observation is the moderate but statistically significant negative correlation between the lobulation index (LI) and both OSI ($r = -0.24$, $p = 0.020$) and RRT ($r = -0.26$, $p = 0.013$). While this trend is statistically supported, it is counterintuitive. One would expect aneurysms with higher LI to be associated with more chaotic and oscillatory flow patterns. A possible explanation is that true daughter sacs—often linked to rupture risk and flow complexity—were relatively rare. And the LI marker as defined here, may be elevated in aneurysms with elongated but smooth morphologies. These cases could exhibit high surface-to-volume ratios without necessarily possessing complex internal flow structures. Therefore, the specificity of LI as a marker may be limited. As this is an preliminary investigation, further studies with larger and more diverse datasets are needed. In contrast, the equivalent radius (ER), which reflects the size of the aneurysm sac based on its volume, showed a

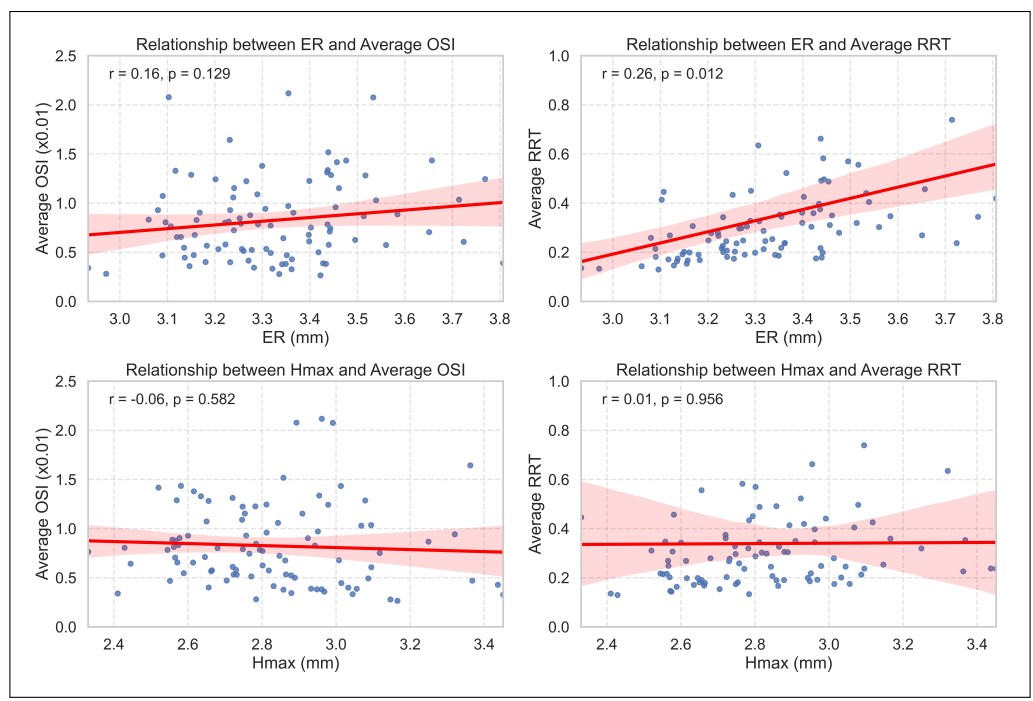

Figure 6: Morphological–biomechanical marker correlations (ER and $H_{max}$)

moderate positive correlation with both OSI ($r = 0.16$, $p = 0.129$) and RRT ($r = 0.26$, $p = 0.012$). This trend is intuitive, as a larger ER corresponds to a larger sac volume, increasing the likelihood of developing slow and recirculating flow. Such regions are associated with disturbed hemodynamics behaviors, including elevated oscillatory shear and prolonged residence time.

## 6 Conclusion

In this paper, we present a new large-scale and open-source dataset designed to support the development of data-driven models for predicting hemodynamics from geometries. The dataset includes 14,000 steady-flow cases and 730 pulsatile-flow cases, each computed using high-resolution CFD simulations with anatomically physiological IA geometries. These geometries model both the aneurysm sacs and their parent vessels as a joint distribution, addressing key limitations in previous datasets that relied on idealized or manually deformed shapes (9; 18).

By leveraging a deep generative model trained on a real IA cohort, we capture a broad range of physiologically plausible geometries representative of real-world anatomical variations. We provide solution data including pressure, velocity, and WSS. Spatial gradients for velocity components and temporal gradient for pressure are also provided. Initial experiments using PointNet and graph U-Nets demonstrate the dataset's utility in enabling WSS pattern prediction, achieving a best relative L2 error of 4.67% on normalized WSS.

We hope this dataset will contribute to the biomechanics and machine learning communities, accelerating the development of neural operators and other data-driven solvers for geometry-conditioned partial differential equations.

## Acknowledgments and Disclosure of Funding

This study was supported by the Imperial College startup funding and MOE-AcRF-Tier1-FRC-FY2024 Grant. We would also like to acknowledge that computational work involved in this study is partly supported by the National University of Singapore's IT Research Computing group under grant number NUSREC-HPC-00001.

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
