# OpenReview forum: "AneuG-Flow: A Large-Scale Synthetic Dataset of Diverse Intracranial Aneurysm Geometries and Hemodynamics"
_NeurIPS.cc/2025/Datasets_and_Benchmarks_Track — NeurIPS 2025 Datasets and Benchmarks Track poster_

### Official Review · Reviewer_JwSJ · 2025-06-30

**Rating:** 5
**Confidence:** 5

**Summary:**

Built by deforming aneurysm templates into synthetic geometries, this dataset offers unparalleled shape diversity and statistical power for morphology‑driven analyses. Each model is paired with CFD‑resolved velocity, pressure, and wall‑shear‑stress fields under steady‑state flow conditions, enabling controlled studies of anatomy–hemodynamic interactions. Eliminating the chronic “small‑sample” bottleneck in intracranial aneurysm research, this large, high‑fidelity resource empowers modern AI and statistical models to develop clinically actionable, individualized risk assessments.

**Dataset Code Accessibility:**

Yes

**Dataset Code Comments:**

The authors provide a fully packaged release of AneuG‑Flow in its final form, with both the 14 000 steady‑flow and 200 transient CFD cases hosted on Zenodo.

**Ethical Considerations:**

No, there are no or only very minor ethics concerns

**Final Justification:**

I support this dataset publication.

**Limitations Weaknesses:**

AneuG‑Flow’s usefulness is tempered by three main drawbacks: first, the dataset includes only 200 transient simulations compared to 14 000 steady‑flow cases, which limits its ability to support models that need rich temporal signals across cardiac cycles; second, every geometry follows a fixed single‑inlet/two‑outlet topology, so methods trained on AneuG‑Flow may struggle to generalize to other clinically relevant branching patterns or more complex vessel networks; and third, boundary conditions are kept constant—patient‑specific variations in flow waveforms and pressure are not sampled—so learned surrogates may underperform when faced with physiological uncertainty, suggesting a need for more diverse pulsatile cases, broader topology classes, and randomized boundary‑condition sweeps in future releases.

**Strengths Contributions:**

AneuG-Flow presents the first open-source, large-scale hemodynamics dataset that combines 14,000 realistic synthetic intracranial aneurysm geometries—generated by a VAE-based shape model trained on clinical cases—with both high-resolution steady-flow and pulsatile CFD simulations, providing volumetric fields of velocity, pressure, wall shear stress, and their spatial gradients. The dataset’s standardized mesh graphs and connectivity enable seamless integration with neural operators and graph learning methods, as demonstrated by baseline experiments achieving a sub-5% relative error in WSS prediction. By vastly expanding the scale and realism of available aneurysm flow data, AneuG-Flow fills a critical gap in geometric-conditioned surrogate modeling and offers a reproducible benchmark for advancing machine-learning methods in neurovascular biomechanics. Its clear documentation, FAIR‐compliant licensing, and comprehensive metadata further ensure that researchers can readily reproduce and extend its use in both academic and clinical research contexts.

---

> ### Author Rebuttal · Authors · 2025-07-28
>
> Thank you for your valuable and constructive comments! We appreciate your recognition of our work's attempt to create a CFD dataset with non-idealized geometries.
>
> ## W1: Regarding the transient case size:
> > The dataset includes only 200 transient simulations compared to 14 000 steady flow cases.
>
> The reviewer is right, 200 transient simulations is a low number. However, this is sufficient for certain deep learning flow network such as MeshGraphNet [1]. Further combined with the 14,000 steady flow cases, strategies of transfer learning from steady flow predictor networks to pulsatile flow predictor networks, which requires smaller number of pulsatile cases, can be tested. Further, by now, we actually have 500 pulsatile cases, and if allowed to, we propose to add them to the dataset after the review phase.
>
> [1] MeshMask: Physics-Based Simulations with Masked Graph Neural Networks.
>
> ## W2: Regarding the limitation of geometry:
> > Every geometry follows a fixed single inlet/two outlet topology, so methods trained on AneuG Flow may struggle to generalize to other clinically relevant branching patterns or more complex vessel networks.
>
> We clarify that AneuG and our current AneuG-flow dataset are intended to only cover 1 specific morphology type, those of bifurcation aneurysm at the Middle Cerebral Artery. We believe that every aneurysm morphology type (bifurcation aneurysm, non-bifurcation aneurysm and, aneurysm on H-shaped vascular structure) need their own generation and flow dataset. We have chosen bifurcation aneurysm morphology type as it’s the second prevalent type of cerebral aneurysm and it’s more challenging than non-bifurcating aneurysms. However, the generation pipeline described in [2] ([19] in the paper) can be extended to other aneurysm types. We have already stated this limitation in section 3.4.
>
> [2] Two-Stage Generative Model for Intracranial Aneurysm Meshes with Morphological Marker Conditioning.
>
> ## W3: Regarding the limitation of boundary conditions:
> > Boundary conditions are kept constant—patient specific variations in flow waveforms and pressure are not sampled—so learned surrogates may underperform when faced with physiological uncertainty.
>
> The reviewer is absolutely right. This is a limitation, and we have already acknowledged it in section 3.4. There is currently a lack of literature on the variations of cranial arterial flow waveforms, as such measurements are not routinely performed in clinical treatment of intracranial aneurysms. And we had found it difficult to incorporate this in our dataset. Nonetheless, we believe that our current dataset is still useful in training network to predict fluid mechanics of flow from a single flow waveform.

---

### Official Review · Reviewer_YXoM · 2025-07-02

**Ethics Flags:** Data privacy, copyright, and consent
**Rating:** 5
**Confidence:** 3

**Summary:**

The paper proposes a large-scale dataset of intracranial aneurysm (IA) geometries and corresponding hemodynamic simulations. The dataset comprises 14,000 steady cases and 200 pulsatile cases. To evaluate the dataset, the authors conduct baseline experiments using PointNet++ and U-Net models for predicting wall shear stress (WSS) maps on the proposed aneurysm dataset.

**Dataset Code Accessibility:**

Partly

**Dataset Code Comments:**

The proposed dataset are set up with conditional access rather than direct access.

**Ethical Comments:**

The paper does not provide details about the 109 real intracranial aneurysm (IA) cases used to train the generative model, which may involve patient privacy or ethical considerations. In addition, the source and generation process of the 200 pulsatile cases are not clearly described.

**Ethical Considerations:**

Yes, there are ethics concerns that require attention by the authors

**Final Justification:**

Thanks for the response, I think it addressed my concerns, therefore I have raised my rating to 5 accept.

**Limitations Weaknesses:**

- In Table 1, it is stated that “14,000 synthetic shapes generated using a generative model trained on a cohort of 109 real shapes.” However, the source and generation method for the additional 200 pulsatile cases are not clearly explained. Additionally, the compared datasets listed in Table 1 should be explicitly named for clarity and transparency.

- In Line 136, the paper mentions that the data generation method follows [19], but this reference appears to be unpublished and has not undergone peer review. This raises concerns regarding its technical credibility. It is unclear whether the generated data has been validated by medical experts.

- Although experimental results are presented in Section 4, it remains uncertain whether these preliminary results meet domain-specific professional or clinical requirements. If improvements are needed, it would be helpful for the paper to provide further discussion or guidance on potential directions for refinement.

- The main paper appears to have sufficient space to include the experimental analyses currently placed in the supplementary materials. Moving relevant content to the main text would enhance the clarity and impact of the paper’s contributions.

**Strengths Contributions:**

- Hemodynamic simulations of aneurysms are of significant clinical importance. The large-scale dataset proposed in the paper, which includes both steady and pulsatile cases, has the potential to greatly support research in this domain.

- The paper provides detailed descriptions of the dataset and experimental setup (Sections 2.3, 3.2, and 4), and includes analysis of the experimental results in the supplementary materials.

---

> ### Author Rebuttal · Authors · 2025-07-28
>
> We thank you for the valuable and insightful comments! Below, we address your concerns and questions individually:
> ## W1: Regarding the setup details of the dataset
> > The source and generation method for the additional 200 pulsatile cases are not clearly explained.
>
> Thank you for raising this issue! We apologize for the oversight. In this work, the geometry configurations used for the pulsatile cases are identical to those for steady cases. Mass flow waveform is imposed at the inlet for pulsatile cases using measurement provided by [1]. Only one flow waveform is used here as there is little literature on the other variants of flow waveforms, and such measurements are not routinely performed in the clinical treatment of intracranial aneurysms. We already stated this as a limitation in section 3.4.
>
> > The compared datasets listed in Table 1 should be explicitly named.
>
> We will revise the paper to add explanations of the pulsatile simulation methodology. Additionally, we will revise Table 1 to explicitly give the names of the datasets.
>
> [1] Geometry of the Carotid Bifurcation Predicts Its Exposure to Disturbed Flow.
>
> ## W2: The shape generator used appears to be unpublished and has not undergone peer review.
> This is a good point, and we agree. In fact, the generator work AneuG [2] ([19] in the paper) has been accepted by **MICCAI 2025**. MICCAI will formally publish the paper in the 1st week of September, in time for the NeurIPS camera ready submission, and we will replace the ArXiv reference with the MICCAI one in that version.
>
> As stated in [2], shapes generated by AneuG has been evaluated by a neuroradiologist, who ranked two versions of AneuG generated cohort the highest in diversity and quality. When using AneuG here, we adopted the version giving the best generation quality. As such, there has been some clinician validation.
>
> [2] Two-Stage Generative Model for Intracranial Aneurysm Meshes with Morphological Marker Conditioning.
>
> ## W3: Although experimental results are presented in Section 4, it remains uncertain whether these preliminary results meet domain-specific professional or clinical requirements.
> We thank the reviewer for this good point. Detailed computations of fluid mechanics parameters have not yet been adopted by clinicians, and remain within the biofluid mechanics research domain currently. However, decades of mechanics and physiological research indicate that such parameters are promising biomarkers for indicating disease progression or deterioration risks (e.g., [3]) . The generation of fluid mechanics ground truth details via CFD simulations is currently a mature area, and can be performed with commercial software to a high quality, such as we have done. Since existing meta-analyses [3] have provided compelling evidence of a correlation between low wall shear stress (WSS) and aneurysm rupture, we chose to predict **WSS vector maps** on the aneurysm pouch as a preliminary experiment.
>
> The preliminary results in section 4 concerns using deep learning for rapid generation of WSS maps on realistic, diverse cranial aneurysm geometries. To date, this has not been achieved, previous work has only applied deep learning in idealized cranial aneurysm geometries (e.g., [4]). DL prediction of WSS has been attempted in other cardiovascular parts, but they are typically trained for a very limited range of geometries [5, 6]. Prediction accuracies for WSS have ranged from 3-5%. In comparison, our error of <5% compares well. However, the reviewer is absolutely right as it remains unclear which biomechanical markers are most effective. For example, emerging metrics derived from fluid–structure interaction (FSI) analyses are currently being explored as potential markers [7]. We will highlight this in the limitations section and expand the discussion on possible directions for future improvement.
>
> [3] Association of wall shear stress with intracranial aneurysm rupture: systematic review and meta-analysis.
> [4] MeshMask: Physics-Based Simulations with Masked Graph Neural Networks.
> [5] Role of physics-informed constraints in real-time estimation of 3D vascular fluid dynamics using multi-case neural network.
> [6] Mesh neural networks for SE(3)-equivariant hemodynamics estimation on the artery wall.
> [7] Fluid structure Interaction analysis for rupture risk assessment in patients with middle cerebral artery aneurysms.
>
> ## W4: Experimental analyses should be moved to the main paper since there’s sufficient space.
> Thank you for this suggestion. We will move the regression analysis results to the main paper for better clarity.
>
> ## Concern regarding the dataset and code accessibility:
> We included the full version of codes as well as links to pre-trained checkpoints in the supplementary material. We apologize for the incomplete GitHub repository. Following the no updates rule during rebuttal, we will update our GitHub repository only after this stage. For the dataset on Hugging Face, we've set the access approval to be automatic.
>
> ## Concern regarding Ethical Considerations:
> > The paper does not provide details about the 109 real intracranial aneurysm (IA) cases used to train the generative model, which may involve patient privacy or ethical considerations.
>
> We did not acquire these cases, but have obtained them from a publicly published dataset: AneuX [8]. Ethics information is already given in the publication. We apologize for missing this in the manuscript, and will revise it to clarify.
>
> [8] Shape trumps size: Image-based morphological analysis reveals that the 3D shape discriminates intracranial aneurysm disease status better than aneurysm size.

---

> > ### Comment · Reviewer_YXoM · 2025-08-04
> > **Thanks for the response**
> >
> > Thanks for the response, it addressed my concerns and I raised my rating.

---

### Official Review · Reviewer_yu5Q · 2025-07-03

**Ethics Flags:** Safety and security
**Rating:** 4
**Confidence:** 2

**Summary:**

This paper introduces AneuG-Flow, a large-scale, high-fidelity CFD dataset of intracranial aneurysm (IA) blood flow dynamics designed to support machine learning research. Based on 427 real IA geometries, the authors synthetically generated over 10,000 3D shapes through controlled deformation and simulated blood flow under eight different conditions. The dataset includes flow parameters, segmentation masks, and geometric data, offering multimodal inputs for diverse tasks. A benchmark for flow estimation is also proposed. The work significantly contributes to data-driven hemodynamic modeling and clinical risk prediction, bridging computational biofluid mechanics with scalable ML applications.

**Dataset Code Accessibility:**

Yes

**Dataset Code Comments:**

It looks good to me.

**Ethical Considerations:**

No, there are no or only very minor ethics concerns

**Final Justification:**

Most of my problems are addressed. I would like to maintain my initial positive score.

**Limitations Weaknesses:**

1. Could the authors provide multiview images and poses for novel view synthesis/ 3D reconstruction?

**Strengths Contributions:**

The paper presents a highly valuable and well-structured dataset for modeling complex cerebral aneurysm hemodynamics.

The use of real clinical geometries and validation by neurosurgeons adds strong credibility and clinical relevance.

The dataset’s scale, multimodal richness, and included benchmark promote reproducibility and broad applicability in ML research.

---

> ### Author Rebuttal · Authors · 2025-07-28
>
> Thank you for your valuable comments! We address your questions and concerns below:
> ## W1: Could the authors provide multiview images and poses for novel view synthesis/ 3D reconstruction?
> Indeed, multi-view images for the 3D geometries will be very informative. We are not allowed to generate any further images or provide further links to external pages at the rebuttal stage. However, **rotational GIFs and detailed mesh images** of both the aneurysm dome and the whole shape can already be found in the original GitHub repository of the shape generator AneuG [1] ([19] in the paper). Moreover, surface mesh files are included in our dataset (Geometries.zip) for users’ convenience.
>
> Nonetheless, we will revise our paper and include more images from different poses, and include rotational gifs for signature shapes in the supplementary material.
>
> To assure the reviewer that the generated shapes are high quality, we noted that according to [19], AneuG generated shapes were evaluated to have excellent generation quality, and has been ranked the best by a neuroradiologist, compared to alternative generators. When using AneuG here, we adopted only configurations giving the best generation quality. Further, although [1] is ArXiv published, it has already been accepted by **MICCAI 2025**, and will be formally published by MICCAI in the 1st week of September, before the camera-ready deadline for NeurIPS. We can replace [1] with the MICCAI reference in the final draft.
>
> [1] Two-Stage Generative Model for Intracranial Aneurysm Meshes with Morphological Marker Conditioning.

---

> > ### Comment · Reviewer_yu5Q · 2025-08-05
> >
> > Thank you for your insightful comments. I recommend that authors try to provide multiview images, camera poses, Ground Truth mesh in the future (as mentioned in the rebuttal). As I am not familiar with this area, I would like to maintain my initial score.

---

### Note · Authors · 2025-08-15

The authors thank all reviewers and Area Chairs for their insightful comments. We summarize the main contributions and limitations of our work in this final remark.

## Strengths mentioned by reviewers:
- Solid clinical grounding, with shapes derived from real cases and validated by neurosurgeons.
- High-resolution volumetric fields of velocity, pressure, wall shear stress, and their spatial gradients.
- Large-scale, standardized graph meshes, and reproducible benchmarks.
- Potential to advance deep learning-based neurovascular biomechanics surrogation.

## Main concerns and our responses:
- Quality of our geometries.
> As stated in [1], shapes generated by AneuG have been evaluated by a neuroradiologist in terms of diversity and quality. As such, there has been some clinician validation. Furthermore, this work has been accepted by MICCAI 2025, and will be formally published by MICCAI in the first week of September. We will replace [1] with the MICCAI reference in the final draft.

- The number of transient cases (200) is relatively small compared to the steady cases (16,000).
> While 200 pulsatile simulations are sufficient for certain models [2] as well as transfer learning strategies, we have expanded to 500 cases and will include them in the final release.

- Boundary condition limitations.
> Our dataset has a few limitations regarding flow boundary conditions, such as a lack of variation in the inlet waveform, primarily because such measurements are not routinely performed in clinical practice. We will emphasize these limitations in the paper.

- Ethics considerations.
> All patients and participants provided written informed consent for the original morphology data. We will clarify the source, the handling of real cases, and the ethics statement of the original patient-specific data in the Methods section.

The authors thank the reviewers and ACs for their constructive feedback. All clarifications and updates mentioned during the rebuttal will be reflected in the final version, ensuring that all raised concerns are addressed.

[1] Two-Stage Generative Model for Intracranial Aneurysm Meshes with Morphological Marker Conditioning.
[2] MeshMask: Physics-Based Simulations with Masked Graph Neural Networks.

---

### Decision · Program_Chairs · 2025-09-18

**Decision:**

Accept (poster)

**Comment:**

First draft to be discussed.

The AneuG-Flow dataset is a large-scale computational fluid dynamics (CFD) resource designed to support machine learning research on intracranial aneurysms. It includes 14,000 steady-flow simulations and 200 pulsatile cases, derived from 427 real patient geometries and expanded into over 10,000 synthetic shapes validated by neurosurgeons. The dataset provides multimodal information (velocity, pressure, wall shear stress, segmentation masks, and geometric data), offers standardized mesh graphs for graph-based learning methods, and proposes benchmarks for flow estimation and wall shear stress prediction.
Documentation, FAIR-compliant licensing, and metadata are included to facilitate reproducibility and reuse.

But this dataset also has limitations :
The number of pulsatile simulations is relatively small compared to steady cases, restricting its utility for tasks requiring temporal flow dynamics.
All geometries follow a fixed single-inlet/two-outlet topology (It applies to the carotid arteries, but carotid aneurysms have a low prevalence), which may limit generalization to more complex vascular anatomies.
Boundary conditions are constant, without patient-specific variability, which reduces physiological realism.
Details on the generation and validation of the dataset—especially for pulsatile cases—are not fully described, and one of the authors' methodological references was unpublished at the time of submission [19].
Ethical aspects, including the handling of real patient cases used to train the generative model, are not fully clarified. Access to the dataset is conditional rather than direct.
Finally, the extent to which the dataset meets domain-specific clinical requirements is not established.